# Sustainable Development Ensued by Social Capital Impacts on Food Insecurity: The Case of Kibera, Nairobi

Emma E. W. Termeer [1,*], Katrine Soma [1], Nina Motovska [1], Oscar Ingasia Ayuya [2], Marvin Kunz [1] and Tinka Koster [1]

1   Wageningen Economic Research, Wageningen University & Research (WUR), Droevendaalsesteeg 4, 6708 PB Wageningen, The Netherlands; katrine.soma@wur.nl (K.S.); nina.motovska@wur.nl (N.M.); marvin.c.kunz@gmail.com (M.K.); tinka.koster@wur.nl (T.K.)
2   Department of Agricultural Economics and Agribusiness Management, Egerton University, Egerton P.O. Box 536-20115, Kenya; ingasiaoa@gmail.com
*   Correspondence: emma.termeer@wur.nl

**Abstract:** The aim of this study is to disclose the social factors of sustainable development goals by exploring the links between three types of social capital (bonding, bridging and linking) and food security in Kibera, an informal settlement located in Nairobi, Kenya. Several studies in the literature have addressed links between food security and social capital. However, a lack of theoretical approaches exist in the literature, which concern the sustainable development theory devoted to urban areas taking into account the sustainable development goals. This study applies a linear regression model on data from 385 households in Kibera to analyze the connection between food security and three types of social capital (bonding, bridging and linking). The results demonstrate that there is a positive impact between our proxies for bonding social capital (cultural diversity and the number of visits to area of origin) and food security. Bridging social capital (measured by trust in strangers) demonstrated a negative impact on food security. Finally, one indicator for linking social capital demonstrated a positive impact on food security (trust in community leaders), whereas the statistical analyses did not find any relationship of the two indicators; 'trust in local politicians' and 'membership of social organisations', with food security. The results demonstrate that insight into social capital can inform the understanding of household food insecurity in vulnerable urban settlements, by illustrating the critical impacts of social drivers in a food system.

**Keywords:** food security; social capital; household survey; informal settlements; Kibera; Nairobi; Kenya

## 1. Introduction

African countries have the highest urbanisation rate globally, as well as an increasingly growing number of people that live in slums. The expectation is that Africa's population will double by 2050, with two-thirds of this increase happening in urban areas [1]. Aside from that, conflicts, unstable economic situations and environmental degradation are connected to rising food insecurity on the continent [2]. In 2015, the Sustainable Development Goals (SDGs) were introduced by the United Nations General Assembly (UN-GA). The scientific literature on sustainable development has since then provided analytical approaches and impact assessments towards the achievements of the 17 different goals and their applicability [3–5]. However, a lack of theoretical approaches exists in the literature, which concern the sustainable development theory devoted to urban areas, taking into account the sustainable development goals (SDGs).

Increasingly, social dimensions are taken into account in understanding the incidence of food insecurity on the household level. These social dimensions are referred to as social capital: the networks and relationships between and within groups of people that they rely on and that complement other forms of capital (e.g., economic or human capital); for example, shared values, norms and trust. A common distinction is made between

three types of social capital: bonding, bridging and linking [6,7]. Putnam [6] defined the distinction between bonding and bridging social capital as 'inward'-looking and 'outward'-looking networks. Bonding social capital can be considered as "personal relations that are based on a sense of collective identity such as family, close friendship and the sharing of the same culture or ethnicity" [8] (p. 3). In contrast, bridging social capital can be considered as "people's relations or links that stretch beyond a shared sense of identity, for example to distant friends, colleagues and associates" [8] (p. 5). Woolcock [7] first defined linking capital as the links between communities and economic, political and social institutions [9]. This latter form of social capital is considered as the relations between people and institutions, for example, reflecting societal status and wealth [8].

To test the interactions between food security and social capital, the Kibera informal settlement in Nairobi provides a suitable context. Although it is referred to as one of the largest slums in Africa, the number of inhabitants in Kibera are highly disputed. Over time, different authors have given varying estimates of the population in Kibera with figures ranging from 170,070 to 700,000, and even up to 1.5 million [10–12].

The aim of this article is to disclose the social factors of sustainable development goals by exploring the links between the three types of social capital (bonding, bridging and linking) and food security in Kibera. The main research question in this survey is: To what extent do bonding, bridging and linking social capital influence food security in slum areas?

## 2. Materials and Methods

### 2.1. Theoretical Frameworks

Several studies demonstrate that social capital can play a role in explaining variation in food security. For example, Martin et al. [13] demonstrated a positive relation between trusting one's neighbour and the lower likelihood of experiencing hunger. Chriest and Niles [14] found a positive relation between high levels of social capital and higher resilience and adaptation capacity to food insecure situations. They also demonstrate that the type of social capital (bridging and bonding) is connected to the way a community uses their physical capital [14]. However, clear evidence of the relation between the three types of social capital and food security is lacking. This knowledge is especially needed in the context of vulnerable settlements, as social capital can take on different forms due to the complexity of informal urban settings [8]. The extent to which the three types of social capital relate to food security in these settings is yet unclear based on the existing evidence.

Slums are the most vulnerable of the informal settlements, and provide very challenging environments to assess the linkages between food security and social capital, as an important aspect of informal urbanism is migration and sometimes conflictual identity politics of urban dwellers, including ethnic affiliations [15] (p. 105). Generally, the residents of urban informal settlement have a higher risk to be food insecure than residents of formal settings, because unexpected events have a greater impact on the food security of these households [16]. Slum populations in Kenya are highly vulnerable to unexpected shocks to their food security situation, as the country is struggling to handle the high rate of rural to urban migration [17] (p. 4).

The Kibera settlement is made up of several villages, with Makina being the largest. The slum population consists of migrants from rural areas that have come to the city for economic opportunities. The residents represent several of the ethnic backgrounds present in Kenya, among which the Luo and Luhya tribes are the highest represented [18]. The fact that most residents are migrants greatly shapes the social dynamics in Kibera, for example, as observed during election times, when different ethnic affiliations and conflicting identity politics come to the fore [15]. Another important characteristic of the settlement is its situation of poverty: estimations are that more than three quarters of households live below the poverty line [19]. This has a significant impact on the food security situations of these households. Nairobi's slums are marked by the lack of important nutrients and unstable incomes of the residents, of which on average 40–50% is spent on food [20].

### 2.2. Methodology of the Research

This study applies a linear regression model on data from a household survey conducted in Kibera in August 2020 among a randomly selected sample of 385 households from twelve villages in Kibera. The data was collected as part of the project 'Feeding cities and migration settlement', which conducts multiple activities in the rural-urban landscape in Kenya, including Kibera. The household survey was developed to serve different project activities and was not designed specifically for this paper.

The survey conducted in Kibera follows a food system approach. The food system approach connects food system activities—production, processing, distribution, preparation and consumption of food—to important biophysical, economic, political and social factors, as well as the outcomes of these activities, such as socioeconomic and environmental dynamics [21]. Assisting in understanding complexities of such systems, including impacts on food security by social activities [22], the food system approach departs from the food system as an interplay of interacting subsystems, such as markets, land and water. The food system approach looks at the impacts of all activities in the system, their interactions and how feedback loops impact socioeconomic and environmental drivers of the system [23,24]. While the food system framework is a great support in investigating complex interactions, it is not possible to analyse everything at every time, and choices must be made. In this survey, the particular focus is on the social driver of social capital as a potential factor influencing food security, thus putting SDG2: Zero hunger up-front.

Figure 1 provides an overview of the food systems approach, defining the multiple factors of outcomes along the 17 SDGs.

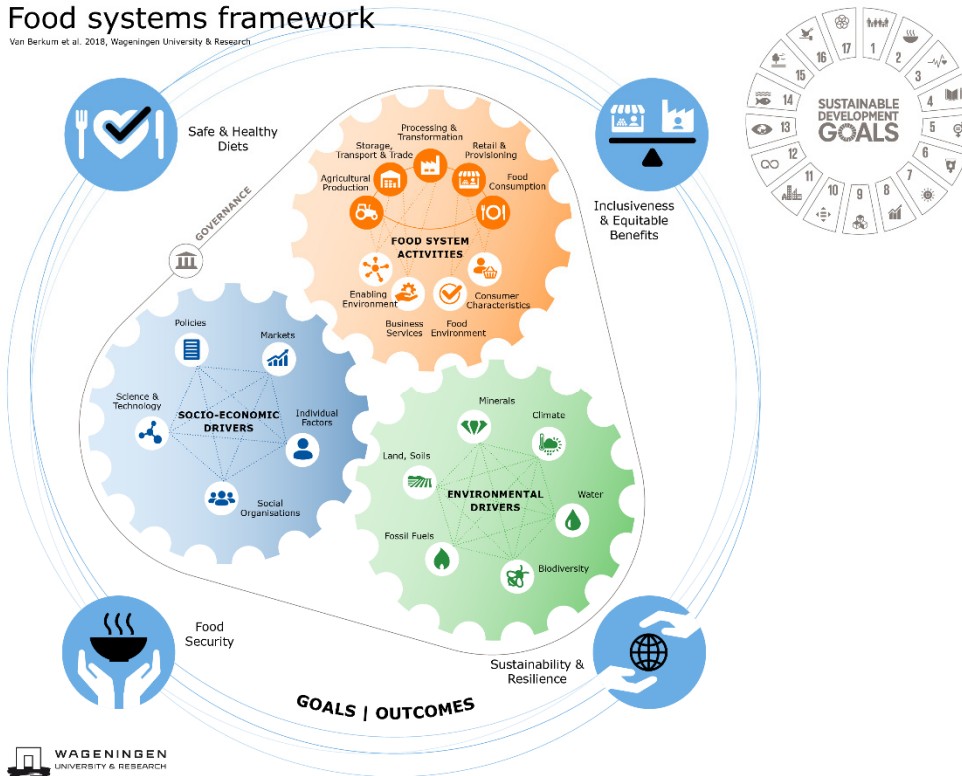

**Figure 1.** A food systems framework with food system activities, the socio-economic and environmental drivers, aiming towards the key outcomes of food security, sustainability and resilience, safe and healthy diets, and inclusiveness and equitable benefits.

Prior to the household survey, a series of in-depth interviews were conducted which provided valuable information, but also raised questions about the ways in which the food system in Kenya functioned. Based on this, a documentary was made (see Supplementary Material: Video S1). This resulted in setting up the design of a new fish value-chain between

Kibera and the Nyeri district, a rural area that supplies small-sized affordable and accessible fish [25]. The new fish value-chain is reported by two documentaries (see Supplementary Materials: Videos S2 and S3). Based on this, the project analysed the fish-food system across the Kibera villages [26], and designed the household survey, of which part of the data is used in this study.

A selection of clusters, households and individuals was made according to a two-stage cluster sample design. The first stage consisted of the clustering of Kibera into thirteen villages. One village was left out of the clustering due to reasons of security on the ground. The other twelve villages were allocated an equal sample. In the second sampling stage, Personal Digital Assistants (PDAs) randomly selected respondents from the target population. This was done by random walk, for which the field supervisor indicated a minimum of 10 important land marks in the cluster per village, such as markets, schools and religious sites. The starting point for the walk was randomly selected by the script, as after which it also selected the direction and sampling interval for the respondents.

Of the 385 respondents, 292 were female and 93 were male. This difference in gender representation can be explained by the availability of females at home during day time, given that men often work long days outside Kibera before returning in the late evening hours. Although the majority of respondents are female, most of them are not the household head, as only 3% of household heads in the sample are female. The respondents' ages ranged from 19 to 59. The mean age was 32 years and the median age 30. The model also tested the link between social capital and food security in the sample of only females and the sample of only males. The results of these analyses can be found in Tables 1 and 2.

This study applies FAO's Household Food Insecurity Access Scale (HFIAS) as a proxy for food security [27]. This scale measures food security by a range of questions related to a household's food consumption patterns. The scale ranges from 0 to 27, with a high score indicating the household experienced high food insecurity. The responses in our dataset range from 0 to 22. To provide a more intuitive scale for this study, a linear transformation was performed to inverse all values by subtracting the maximum value of 22 from all observations, which was then multiplied by $-1$. This means that the more food secure the household is, the higher the score. Such a linear transformation also greatly improves interpretability of the available data. The scale can be categorised into four groups. An overview of the shares of the sample categorised into different levels of food security is provided in Figure 2.

**Table 1.** Regression results applying the multivariate regression model (1) in a total of seven different runs.

| | Run 1 | | Run 2 | | Run 3 | | Run 4 | | Run 5 | | Run 6 | | Run 7 | |
|---|---|---|---|---|---|---|---|---|---|---|---|---|---|---|
| | Coef. | Std. Err. | Coef. | Std. Err. | Coef. | Std. Err. | Coef. | Std. Err. | Coef. | Std. Err. | Coef. | Std. Err. | Coef. | Std. Err. |
| Cultural diversity | 0.6649 * | 0.32 | | | | | | | | | | | 0.7058 | 0.32 |
| Trust strangers | | | −0.7403 * | 0.35 | | | | | | | | | −0.9927 ** | 0.35 |
| Origin visits | | | | | 0.9417 * | 0.43 | | | | | | | 1.0169 * | 0.43 |
| Trust community leaders | | | | | | | 0.8578 * | 0.35 | | | | | 0.9055 * | 0.36 |
| Trust local politicians | | | | | | | | | 0.2369 | 0.33 | | | 0.2715 | 0.34 |
| Membership of social organisations | | | | | | | | | | | −0.2527 | 0.35 | −0.2613 | 0.33 |
| Household size | −1.0702 ** | 0.4 | −1.0774 ** | 0.4 | −1.0711 ** | 0.4 | −1.1195 ** | 0.39 | −1.0929 ** | 0.40 | −1.0699 ** | 0.41 | −1.0314 ** | 0.39 |
| Age household head | −0.4183 | 0.51 | −0.2759 | 0.51 | −0.4366 | 0.51 | −0.1416 | 0.52 | −0.3375 | 0.52 | −0.3106 | 0.52 | −0.1373 | 0.51 |
| Education level household head | 0.5526 | 0.47 | 0.6607 | 0.46 | 0.7118 | 0.46 | 0.9643 * | 0.46 | 0.7615 | 0.46 | 0.7788 | 0.47 | 0.6181 | 0.46 |
| Female household head | 0.2517 | 0.36 | 0.1821 | 0.36 | 0.2756 | 0.36 | 0.2543 | 0.36 | 0.2033 | 0.36 | 0.1632 | 0.37 | 0.3489 | 0.35 |
| Years in Kibera | 0.4860 | 0.45 | 0.4374 | 0.45 | 0.6016 | 0.46 | 0.4838 | 0.45 | 0.4596 | 0.46 | 0.4840 | 0.46 | 0.5811 | 0.44 |
| Intercept | 14.4750 *** | 0.35 | 14.5434 *** | 0.35 | 14.5701 *** | 0.35 | 14.5612 *** | 0.35 | 14.4793 *** | 0.36 | 14.5177 *** | 0.36 | 14.6568 *** | 0.35 |
| $R^2$ | 0.071 | | 0.072 | | 0.073 | | 0.079 | | 0.052 | | 0.052 | | 0.144 | |
| F for change in $R^2$ | 3.328 * | | 3.375 * | | 3.422 * | | 3.623 * | | 2.67 * | | 2.67 * | | 3.779 *** | |

* Some significant, ** Moderately significant, *** Highly significant.

**Table 2.** Regression results applying the multivariate regression model (1) in a total of seven different runs disaggregated for the female and male sample.

| Female Sample | Run 1 | | Run 2 | | Run 3 | | Run 4 | | Run 5 | | Run 6 | | Run 7 | |
|---|---|---|---|---|---|---|---|---|---|---|---|---|---|---|
| | Coef. | Std. Err. | Coef. | Std. Err. | Coef. | Std. Err. | Coef. | Std. Err. | Coef. | Std. Err. | Coef. | Std. Err. | Coef. | Std. Err. |
| Cultural diversity | 0.6397 | 0.34 | | | | | | | | | | | 0.6477 | 0.33 |
| Trust strangers | | | −0.7594 * | 0.36 | | | | | | | | | −1.0153 ** | 0.35 |
| Origin visits | | | | | 0.9742 * | 0.38 | | | | | | | 0.9989 ** | 0.38 |
| Trust community leaders | | | | | | | 0.9168 * | 0.37 | | | | | 0.8738 * | 0.38 |
| Trust local politicians | | | | | | | | | 0.3922 | 0.35 | | | 0.3677 | 0.36 |
| Membership of social organisations | | | | | | | | | | | −0.2912 | 0.36 | −0.2849 | 0.34 |
| Household size | −1.0965 ** | 0.4 | −1.1116 ** | 0.42 | −1.0862 ** | 0.4 | −1.1459 ** | 0.4 | −1.1165 ** | 0.41 | −1.1102 ** | 0.41 | −1.0423 ** | 0.38 |
| Age household head | −0.5494 | 0.52 | −0.4341 | 0.52 | −0.6131 | 0.51 | −0.206 | 0.52 | −0.4423 | 0.52 | −0.4258 | 0.52 | −0.2943 | 0.56 |
| Education level household head | 0.1222 | 0.48 | 0.1846 | 0.48 | 0.2226 | 0.47 | 0.5588 | 0.48 | 0.295 | 0.48 | 0.3297 | 0.48 | 0.1824 | 0.48 |
| Female household head | 0.0606 | 0.38 | −0.0637 | 0.38 | 0.0791 | 0.38 | 0.0787 | 0.38 | 0.003 | 0.38 | −0.0502 | 0.39 | 0.1785 | 0.37 |
| Years in Kibera | 0.3291 | 0.6 | 0.3061 | 0.49 | 0.4852 | 0.5 | 0.2929 | 0.49 | 0.2659 | 0.5 | 0.3411 | 0.5 | 0.4305 | 0.48 |
| Intercept | 14.2412 *** | 0.36 | 14.2673 *** | 0.35 | 14.1782 *** | 0.35 | 14.2657 *** | 0.36 | 14.2039 *** | 0.37 | 14.2348 *** | 0.37 | 14.2915 *** | 0.35 |
| $R^2$ | 0.099 | | 0.104 | | 0.115 | | 0.112 | | 0.086 | | 0.083 | | 0.21 | |
| F for change in $R^2$ | 2.937 ** | | 3.112 ** | | 3.481 ** | | 3.392 ** | | 2.532 * | | 2.43 * | | 3.775 *** | |

**Table 2.** *Cont*.

| Male sample | Run 1 | | Run 2 | | Run 3 | | Run 4 | | Run 5 | | Run 6 | | Run 7 | |
|---|---|---|---|---|---|---|---|---|---|---|---|---|---|---|
| | Coef. | Std. Err. | Coef. | Std. Err. | Coef. | Std. Err. | Coef. | Std. Err. | Coef. | Std. Err. | Coef. | Std. Err. | Coef. | Std. Err. |
| Cultural diversity | −0.0252 | 1.03 | | | | | | | | | | | −1.5409 | 2.77 |
| Trust strangers | | | −0.6567 | 1.99 | | | | | | | | | 3.3739 | 4.62 |
| Origin visits | | | | | −1.5110 | 4.14 | | | | | | | −4.3820 | 7.34 |
| Trust community leaders | | | | | | | −0.3198 | 1.15 | | | | | −1.5153 | 2.07 |
| Trust local politicians | | | | | | | | | −0.4653 | 1.46 | | | −3.1099 | 4.75 |
| Membership of social organisations | | | | | | | | | | | 3.136 | 1.36 | 3.1405 | 2.21 |
| Household size | −0.9883 | 1.2 | −1.1563 | 2.04 | −0.2823 | 2.77 | −1.1223 | 2.04 | −1.2073 | 2.09 | −4.310 | 2.11 | −3.2738 | 4.5 |
| Age household head | −0.792 | 2.6 | −0.3278 | 2.89 | −1.7678 | 3.71 | −0.5855 | 2.64 | −0.2791 | 2.99 | 1.023 | 2.16 | −0.5749 | 5.75 |
| Education level household head | 4.3081 | 2.13 | 4.2449 | 2.05 | 4.6279 | 2.23 | 4.2483 | 2.05 | 3.6798 | 2.81 | 3.953 * | 1.6 | 1.6981 | 4.85 |
| Female household head | 1.2796 | 1.33 | 1.4649 | 1.43 | 1.202 | 1.32 | 1.3153 | 1.32 | 1.4272 | 1.39 | 2.986 * | 1.27 | 3.3019 | 2.14 |
| Years in Kibera | 2.4822 | 0.6 | 2.3374 | 1.45 | 2.7288 | 1.55 | 2.3737 | 1.44 | 2.1377 | 1.75 | 2.577 * | 1.09 | 1.5828 | 2.89 |
| Intercept | 16.2596 *** | 1.69 | 16.1999 *** | 1.66 | 15.0154 ** | 3.77 | 16.292 *** | 1.66 | 16.697 *** | 2.17 | 17.442 *** | 1.39 | 17.7314 *** | 5.28 |
| $R^2$ | 0.545 | | 0.551 | | 0.553 | | 0.55 | | 0.551 | | 0.726 | | 0.78 | |
| F for change in $R^2$ | 1.599 | | 1.638 | | 1.647 | | 1.627 | | 1.636 | | 3.53 * | | 0.95 | |

* Some significant, ** Moderately significant, *** Highly significant.

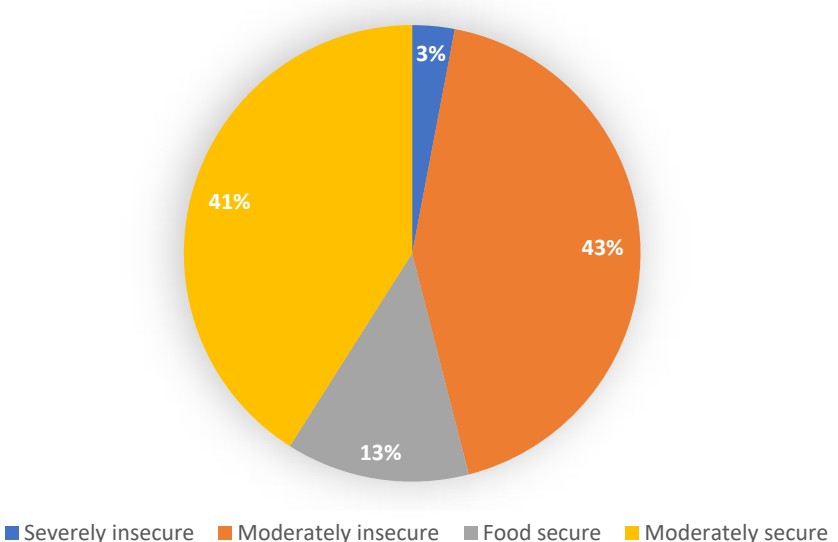

**Figure 2.** Food security categories (*n* = 385).

The household survey was not designed to measure social capital. However, the survey included several variables that represent the subtypes of social capital (bonding, bridging and linking) (Figure 3). These variables include cultural diversity and visits to area of origin (for bonding social capital), trust in strangers (for bridging social capital), and trust in local politicians, trust in community leaders and membership of social organisations (for linking social capital).

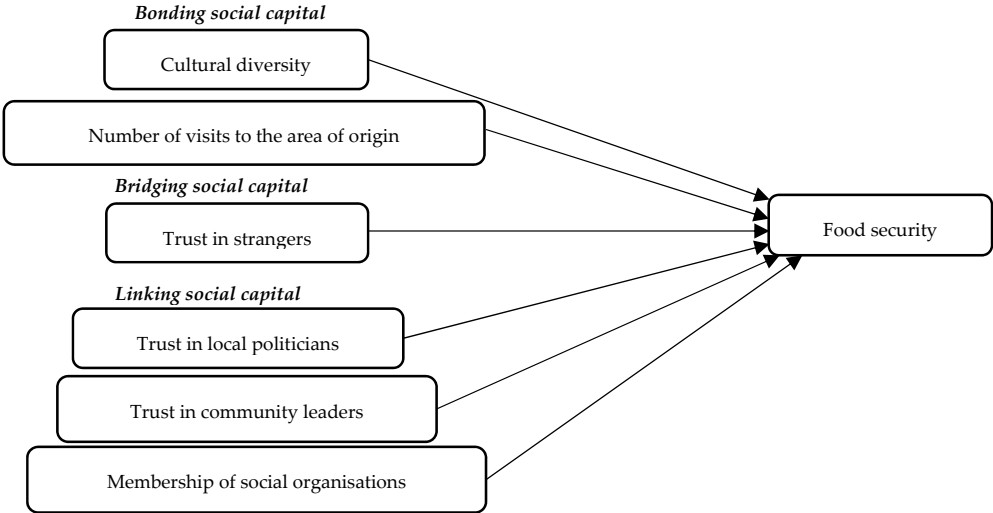

**Figure 3.** Independent variables of social capital (based on a literature review, inspired by among others [6–9]).

Figure 3 shows the independent variables used as proxies for the different types of social capital. The first variable used to measure bonding social capital was the number of visits to the area of origin. Slums are populated by a large number of rural to urban migrants, who often rely on the assistance of their families in situations of (food) insecurity [28]. The number of visits to the area of origin therefore provides a proxy measuring these connections and their strength. This was measured by the question: "How often do you visit your rural areas in a year?". On average, respondents visited their rural area 2.51 times a year. After removing the outliers, the number of visits ranged from 1 to 9. Although the number of visits might also depend on the distance one has to cover, social ties arguably depend more on the actual number of visits than the physical distance. The second variable

used to measure bonding social capital is cultural diversity, measured by the question: "What percentage of your neighbours share your same cultural practices?". This variable was used to provide an indication of shared collective identity. On average, the percentage of neighbours sharing cultural practices was 45% and ranged from 0 to 100.

For bridging and linking social capital, trust in relation to different groups was assessed. To investigate whether social ties to people outside of those with a shared identity can explain household food security, the variable called trusting strangers was used. To measure the relations between people or groups and societal institutions, trust in community leaders, trust in local politicians and membership of social organisations were used. For the latter, 58% of the respondents in our sample indicated to have a household member with formal or informal membership of a social group. The levels of trust in the sample ranged from very low to very high. The following section presents the results of our analysis.

## 3. Results

### 3.1. The Developed Model

This study applies a linear regression model to measure the relation between proxies of social capital types and food security. The regression analysis allows for increased understanding of the extent to which the chosen variables can explain variance in the food security score. Control variables were used to be able to attribute the variance in food security to the social capital variables. These control variables were household size, age of the household head, education of the household head, gender of the household head and the years the household has spent in Kibera.

The outliers were removed for the independent variables, which were checked for normality and standardised. The model was run seven times, to arrive at the estimates of the main model analysing the relation between social capital types and food security, including parts of, or the complete formula presented below (1). The runs 1–6 of the model included only one of the following variables at the time, respectively; cultural diversity ($\beta_1$), trust strangers ($\beta_2$), origin visits ($\beta_3$), trust community leaders ($\beta_4$), trust local politicians ($\beta_5$) and membership of social organisations ($\beta_6$). In the last run, all variables, as listed in (1), were included:

$$
\begin{aligned}
\text{Food security score} \\
= \beta_0 + \beta_1 * \text{Cultural diversity} + \beta_2 * \text{Trust strangers} + \beta_3 \\
* \text{Origin visits} + \beta_4 * \text{Trust community leaders} + \beta_5 \\
* \text{Membership of social organisations} + \beta_6 \\
* \text{Trust local politicians} + \beta_7 * \text{Household size} + \beta_8 \\
* \text{Household head age} + \beta_9 * \text{Household head education} \\
+ \beta_{10} * \text{Household head gender} + \beta_{11} * \text{Years in Kibera}
\end{aligned}
\tag{1}
$$

The independent predictors had low levels of correlation, due to which multicollinearity was considered to be non-problematic in this case. Three households were removed from the dataset due to potentially faulty or inconsistent responses that skewed the data considerably. One of these households was excluded from the sample due to an incorrectly reported food security score. Another household showed an exceptionally high number of visits to the area of origin that could not be explained by any other data points. The third household was considered an outlier as it displayed extreme responses.

### 3.2. Testing of the Model

The results of the regression model assessments, conducted to improve our understanding of the effects of the predictors on food security, are presented in Table 1. The results segregated by gender (male and female samples) are presented in Table 2. This includes a total of seven runs, including different combinations of the variables in the regression model (1), as described in the previous section. The multiple linear regression could explain around 14.5% of the variance in food security (an adjusted $R^2$ of 0.144). This

is statistically significant and means that a relation exists between the chosen social capital variables and food security. Although this $R^2$ may seem relatively low, this is expected because of the nature of the analysis. The use of an existing dataset including proxies for social capital will never fully explain the variation in the data. However, this $R^2$ shows that a significant part of the variation can be explained by the model. It cannot be determined whether the relation is causal or correlational, as the data source does not grant such a conclusion. In the interpretation of the results in what follows, it is important to take into account that food security is the dependent variable. As such, a positive coefficient means a higher score on the independent variable and relates to higher food security.

In the assessment with all indicators of social capital (i.e., run 7), the results confirm the outcomes of the assessments conducted with only two variables at the time (i.e., runs 1–6). The results are not disturbed by a potentially problematic level of collinearity, as all Variance Inflation Factor (VIF) scores are below a level of 2. According to Garc*í*a et al. [29], only VIF scores over 4 can be considered problematic. The linear relations between co-ethnic and other predictors are not high overall. The correlation between cultural diversity and trust in the community is significant, but at −0.1 too low to confirm any relationship.

Four of the six variables used as proxies for social capital that were included as independent variables in runs 1–6 were significant. The variables of cultural diversity, visits to the area of origin and trust in community leaders were significant with a positive linear relation with food security in the hypothesised direction. Trust in strangers was found to be inversely related with food security, meaning that households that have a high level of trust in strangers are on average less food secure. No significant relations between food security and membership of social organisations or trust in local politicians was found. Trust in local community leaders show a significant relation with food security, meaning households with higher levels of trust are on average more food secure.

To measure bonding social capital, the variables of cultural diversity and number of visits to area of origin were used. For cultural diversity, the results indicate a significant and positive relation with food security with an effect size smaller than 1. This means that households that indicate their neighbours do not share the same cultural practices are on average more food secure. For the number of visits to the area of origin, the results demonstrate that households that visit their area of origin more frequently are on average more food secure. The effect of visits to a rural area is slightly greater than the effect of cultural diversity. This is in line with the theory that suggests households that have high levels of bonding social capital can rely on their network to access food during crises [8].

In summary, the results demonstrate that access to food is higher for people living in Kibera with a large social network, and with tight bonding with rural communities and relatives. Moreover, people with shared ethnic backgrounds will more likely ensure that households are more food secure by sharing food. Further, increased food security can be ensured by cultural diversity in a neighbourhood, and also, food security is enhanced when trust relations with the community leader in the slum is increasing. Food security is demonstrated to decrease when peoples' trust in strangers increase, indicating vulnerable situations for these people who may lack other closer relationships. Food security has not shown to increase as a result of increased trust relations with local politicians, or with membership in organisations. The results have been estimated by means of a linear regression model for which correlations between the variables have been tested by means of statistical tests. The reported results apply to both the entire sample and the sample for female respondents, but looking at the male sample separately does not yield any significant results. This can be explained by the large share of female respondents (76%), and thus a relatively small sample of male respondents. Accordingly, this study cannot conclude on the male household heads who responded to the survey, because of the absence of significant results in the sample of only male respondents (Table 2).

## 4. The Implications of the Results

The existing sustainability literature highly focuses on food security aiming at SDG2; zero hunger [3–5]. Food security is defined by the FAO [30] (p. 1) as a situation "when all people, at all times, have physical and economic access to sufficient, safe and nutritious food". This reflects four fundamentals of food security: food access, food utilisation, food availability, and the stability in access to food. The level of food security is linked with sustainability levels of other SDGs, such as SDG1: No poverty, SDG3: Decent health and well-being, SDG8: Decent work and economic growth; and SDG13: Climate action. With low resiliency, communities can be confronted with dramatic changes in relatively short periods of time. These mainly happen in crisis events, such as economic, health and environmental crises, or their combination [30]. Given that these events of crises and instability are expected to increase in the coming years, due to population growth, increasing urbanisation and climate change, the challenge of food security, especially in areas hardest hit by these changes, are expected to become even greater [31]. Food insecurity has an effect on long-term well-being beyond the negative consequences for health. Some of these are possible developmental issues, worsened school performance of children [32], development of psychological issues such as depression and anxiety [33], and even increase in social tensions [34]. Over the years, increasing attention has been paid to the context of social dimensions in food security issues to explain how social relations at the community level affect household coping strategies in food insecure situations [35]. The concept of social capital is used to gain insight into the social dimensions and define its place within the sustainability literature.

The relationship between food security and levels of social capital has been touched upon by several studies. The study by Martin et al. [13] was one of the first to make an explicit link, demonstrating that social capital at both the community and household level associated significantly with household food security among low-income households in the US. Of the chosen indicators for social capital, reciprocity among neighbours appeared as a particularly large contributor. The link has also been studied in African contexts, mostly to determine the effect on food security in rural areas. Dzanja et al. [36] found social capital has a positive influence on food security in Malawi, although the effects vary depending on the form of social capital: group memberships, informal networks and variables associated with trust and participation all improved the food security of households. Sseguya et al. [37] found similar results in Uganda, as households with bridging and linking social capital tended to be more food secure, as well as households with social capital in the form of cognitive bonding (such as awareness of generalised norms and mutual trust). A recent study by Niles et al. [38] combined data from 11 low-income countries in East and West Africa and South Asia and found that multiple scales of social capital, both within and outside the household, correlate with household food security, although they point out that there are important interactions to take into account between these scales. For example, household group membership overall is correlated with better food security outcomes, but the type of group and way of collaboration correlate with the outcomes, suggesting that the context of these participatory indicators matters [38]. Gallaher et al. [11] studied the link between urban agriculture, social capital and household food security in Kibera. Their results demonstrate that households participating in urban agriculture have higher levels of social capital, measured by the exchange of goods and services between households and the quality of the relationships with their urban neighbours. This in turn has a positive impact on the food security situation of households.

Lack of trust can increase vulnerability, depending on the studied situation, indicating the level of risk [39]. Contextual characteristics are essential to explain differing findings across different surveys in the literature. Based on the outcomes of this survey, trust has been demonstrated to play a vital role in understanding relationships between social capital and food security in Kibera. By investigating social capital proxies, and the categories of the *bonding, bridging and linking* effects of social capital, the results demonstrate that trust must be understood within this specific context to understand how it relates with food security.

First, in Kibera, the *bonding* dimension of social capital, investigated as 'cultural diversity' and 'number of visits to area of origin', has been demonstrated to clearly have a link with food security. Cultural diversity is identified by the question: What percentage of your neighbours share your same cultural practices? With larger cultural diversity, the food security increases. This is explained by the observation that diversity of culture implies diversity of backgrounds which comes with acceptability of meals commonly consumed within other communities living close by in Kibera. Because of a so-called 'sharing of meals culture', they can taste other communities' dishes, as well as observe how to prepare and consume, which makes it easy to change diets in times of need. The acceptability of food from other communities means access to diverse food in the market, which provides opportunities for bringing different products into the market (e.g., beans, fish, flour, maize, indigenous vegetables, etc.). If one of the products are lacking, they can more easily replaced this gap with other products, brought in from other cultural backgrounds. Note that when food shortage exists, it is common to fill gaps by means of borrowing food [40].

The question asked for 'number of visits to area of origin' is: How often do you visit your rural areas in a year? It is not surprising that visits to their own rural area increases food security, given that agricultural products are available on the farms. During the visits, households carry food from the rural areas to supplement their purchases in the informal urban settlements from their own rural farm or family production, in some cases without payments or through purchases at relatively low prices. Higher income households visit relatives more often, implying that the lower income groups not only have less opportunities to purchase food in Kibera, but have less opportunities to access food. In the literature, food borrowing from relatives, as a common coping strategy for food security, has been investigated more often across various countries [40–43]. Zingel et al. [43] explained that families to people living in informal settlements probably serve as the most important source of credit, whether financially or in-kind borrowing.

Moreover, borrowing food from friends and neighbours is a commonly employed practice [41]. During Covid-19, one strategy to combat its spread was for people to stay at home due to the order issued by the government, resulting in restricted access to food by households. The social capital created over time with friends and relatives provided a pathway for food insurance, because friends and relatives dependent on each other exchanged food provision based on the prevailing food conditions in their households. The sharing of food in informal settlements is common to bridge the food gap and is dependent on the social ties created with neighbours and relatives.

Second, looking at the proxy for *bridging* social capital, the variable 'trust in strangers' showed a negative influence on food security, which can be explained by the choice of trusting strangers mostly when closer relatives or institutions are not available. Such situations can indicate a vulnerable state of life in Kenya. This implies that when people trust strangers, they are likely to be more vulnerable, which is a situation with increased food insecurity. In this case, the risky relation of trusting a stranger might be the necessity for those that do not have close family or friendships nearby. Low trust can also be an indication of proneness to conflict. Unstable situations threaten household resilience and generally lead to more food insecurity [44].

Third, 'trust in community leaders' was an indicator for the *linking* as a proxy for social capital, which is demonstrated to have a positive impact on food security in this survey. This is explained by the important role these informal leaders have in their assigned neighbourhood of responsibility, an assignment which often is informal. In each of the 14 villages in Kibera, a community leader takes responsibilities, particularly the children and the women, to ensure their safety, facilitate social gatherings, assist sick people and link the labour force with employment opportunities. For instance, in the village called Lindi, the community leader works closely with the policy in a project called 'drop the gun', and given trust relations among youngsters with him, guns are abolished that the police would never ban otherwise. The indifference to 'trust in local politicians' shown in the results of this survey as an explanatory factor to food security is not surprising, because Kibera is

an informal settlement where politicians have low influence. The variable 'membership of social organisations' was not found to have a relation with food security. This makes sense in the context of Kibera, where people are less formally organised. Nevertheless, in Kibera, the women mostly take part in frequent informal gatherings, which most likely lead to enhanced trust relations and also increased food security by close relationships offering help in times of need. A study in Uganda revealed that similar women's group are increasing their access to microfinance as well as access to food in case of need [45]. This points to the fact that informal settlements hold a series of informal social interactions and gatherings, which are increasing trust relations, and in different ways enhancing food security.

In Kibera, a series of coping strategies to food insecurity have been observed. For instance, households in food secure situations tend to reduce food consumption and buy street or precooked foods, which are cheaper. This has also been observed by [41]. Another common coping strategy of households in food insecure situations is to purchase food on credit or loans [46]. Notably, borrowing money through informal transactions may result in inability for repayments, rendering people to even more food insecure situations. Furthermore, one of the coping strategies to food insecurity is sack gardening. It is common in Kibera to cultivate vegetables by means of a sack (not plastic because this is banned in Kenya, but made by an organic material), laid down on the ground filled with soil, and with some openings on top allowing the plants to grow. A study looking into the effects of sack gardening on food security in Kibera concluded that even if the effect is rather limited, it allows the ability to have an unhindered access to food sources in case of an absolute need [47].

Applying a food system approach is about capturing the complexities of a system using a holistic approach. By analysing the relationships between social factors such as social capital (bonding, bridging and linking) and food security, this survey has addressed important relationships not often brought up from in the literature. Still, the survey does not capture all factors and feedback loops in such a system. While we would be interested in documenting the whole food system dynamics, this is never an opportunity given the complexities of such systems. The household survey is designed to focus on a limited, although very interesting, set of relationships (see Table 1). The dataset applied is limited to one round of interviews, which could be increased to more regularity of data selection, for instance, on a yearly basis. Moreover, the number of interviewees are 30–35 per village, across 12 villages, and could be increased in sample size. Relevant feedback loops could involve influencing factors related with migration patterns, gender inequalities, food items consumed, income levels and employment.

The findings suggest that upcoming projects should explicitly consider social capital and trust relationships within the system, because they are critical factors explaining why projects may fail or succeed. Notably, the social factors of bonding, bridging and linking operate independently of formality in the economy and settlements. It is recommended in future research to document how the informal economy may be strengthened or banished across the social capital dimensions. Informality as such is covering highly criminal sectors, as well as communities with strong trust relationships, and clarity of different categories needs to be defined in greater detail. Based on this survey, the community structures stemming from pre-colonial times are of high interest to investigate in future research. Moreover, relevant topics in future research will also include: 1) Exploration of trust relationship across businesses, communities and public sectors comparing successful and failing projects aiming at enhancing food systems towards the SDGs, and 2) investigation of digitisation developments through the food system in ways that can enhance affordability and accessibility of nutritious food for the inhabitants of ever-growing informal settlements in urban areas in future.

## 5. Conclusions

The scientific novelty of this research, disclosing the social factors of sustainable development goals by exploring the links between the three types of social capital (bonding, bridging and linking) and food security in Kibera, can be observed across four levels. First, this research provides new and valuable insights to the sustainability literature. To reach the SDGs, including SDG1: No poverty, SDG3: Decent health and well-being, SDG8: Decent work and economic growth; and SDG13: Climate action, a food system approach must be reached. A food system approach applies a holistic approach to capture the complexities of a system. The important relationships between social factors such as social capital (bonding, bridging and linking) and food security are analysed in this survey, and contribute to further deepening of our understanding of sustainability theories for urban settings with informal settlements.

Second, given that social capital has been criticised for operating as a container concept, in this survey a total of three categories have been carefully specified for trust relations across interactions, in terms of bonding, bridging and linking [6,7]. A series of indicators have been analysed statistically within these categories, and the results obtained are contributing to increasing acknowledgments of informal settlements with potentially high social capital mechanisms, which can contribute to increase food security.

Third, this survey shows that for people living in informal settlements, such as Kibera, a large social network and tight bonding with rural communities and relatives provides more opportunities to access food. More precisely, our results demonstrate that, on the one hand, people with shared ethnic backgrounds are more likely to share food, making households more food secure, and on the other hand, cultural diversity in a neighbourhood can increase food security by bringing in different products from different places, with increased opportunities to fill gaps when, e.g., the fish supply to Kibera is low. Besides the neighbourhood in informal settlements and the relatives in rural areas, trust relations with the community leader in the slum contributes, with enhanced food security. This explains the important role that the community leads have in supporting their neighbourhood with enhanced wellbeing. Food security has not been demonstrated to increase as a result of increased trust relations with local politicians, or with membership in organisations, which may imply that informality is not easily replaceable by formality.

Fourth, this survey has triggered the interest and curiosity of the informal sector worldwide and its role in enhancing food security and employment opportunities in the food system among the very low-income groups. The informal economy could be considered as comprised of all forms of 'informal employment'—that is, employment without labour or social protection—both inside and outside informal enterprises, including both self-employment in small unregistered enterprises and wage employment in unprotected jobs [48]. However, this is only one of several existing definitions of the informal economy. In upcoming research, it is recommended to emphasise the informal settlements and their potentials and critical role in strengthening transitions towards a more food secure society in future—with zero hunger for all (SDG2).

Finally, research on social capital can be advanced in upcoming research on social drivers within a food system and specification of impacts on different livelihood factors. For instance, it would be of interest to investigate more in detail the social capital and food security relations across genders. Moreover, examples that can inspire for investigating social capital within food systems can be based on, among others: (1) the buyer–supplier relationship literature, where antecedents, benefits, risks, and boundary conditions have recently been investigated [49], (2) the literature investigating extents to which social capital can contribute to negative societal impacts, for instance, for political violence [50] or (3) the literature investigating social capital as an explanatory factor to discrepancies of economic growths across regions [51].

**Supplementary Materials:** Video S1: DOCUMENTARY of Kibera (Kibra) documentary: https://www.youtube.com/watch?v=K_goJu2encg&t=7s; Video S2: DOCUMENTARY of Nyeri-Kibera

food system: https://www.youtube.com/watch?v=b4oGoYuCnJ0; Video S3: DOCUMENTARY of Nyeri fish farm documentary: https://www.youtube.com/watch?v=2MYOUZdjKVs&t=1s.

**Author Contributions:** A draft of this paper was written by E.E.W.T. and N.M., and compiled by K.S., with inputs from all co-authors. Conceptualization, E.E.W.T.; Data curation, O.I.A., M.K. and T.K.; Investigation, N.M.; E.E.W.T. and T.K.; Supervision, K.S. All authors were involved in the research design and paper conceptualisation. All authors have read and agreed to the published version of the manuscript.

**Funding:** This research was carried out within the motif 'Feeding cities and migration settlements' (2282700540), as part of the programme of Food Security and Valuing Water (KB-35-002-001) of Wageningen University and Research and was subsidised by the Dutch Ministry of Agriculture, Nature and Food Quality.

**Institutional Review Board Statement:** Ethical review and approval were waived for this study, based on the Research Ethical guidelines of Egerton University in Kenya, who conducted all the interviews.

**Informed Consent Statement:** Informed consent was obtained from all subjects involved in the study.

**Data Availability Statement:** The original dataset is not public at this stage, following the Egerton University data policy in Kenya.

**Acknowledgments:** The authors would like to acknowledge the Wageningen University and Research (WUR) Programme on "Food Security and Valuing Water" that is supported by the Dutch Ministry of Agriculture, Nature and Food Security, in a project called "Feeding cities and migration settlements" (2282700540). Special appreciation goes to the attendances of the workshop held in Nairobi on 29 May 2020, and the interviewees in Kibera during visit in December 2019. Special thanks go to Benson Obwanga (Laikipia University) and Gabriel Francis Mwangi, who invited us to Kibera, and supported us to organise the workshop and preparatory interviews. Additionally, a very special thanks to all the Nyeri stakeholders who contributed to setting up the new fish value chain, including the fish farmers.

**Conflicts of Interest:** The authors declare no conflict of interest.

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
