# Peer review of "Sustainable Development Ensued by Social Capital Impacts on Food Insecurity: The Case of Kibera, Nairobi"

_sustainability, doi:10.3390/su14095504_

Round 1

Reviewer 1 Report

Dear Sir,

The paper attempts to examine “Social capital – an explanatory factor to food security? Results 2 from a survey conducted in Kibera, Nairobi”. After reviewing, I find that this paper is interesting. For better contribution to the literature, I have some revisions that are good for enhancing the quality of the manuscript.

Author Response

Dear Reviewer 1,

The authors would like to thank Reviewer 1 for the good comments that have assisted us to improve this manuscript. We had a professional English reviewer editing the English language of the manuscript. The comments are addressed in the article as follows:

Comment by Reviewer 1

Author explanation of how to improve

1. The novelty of this paper should be further discussed in the section of introduction.

Thank you for this comment. The novelty of this study is now better described in the Introduction, see for instance: ‘In Africa, the increasing food insecurity is predominantly attributed to conflict, unstable economies, and environmental degradation [26]. In 2015, the Sustainable Development Goals (SDGs) were introduced by the United Nations General Assembly (UN-GA). The scientific literature on sustainable development has since then provided analytical ap-proaches and impact assessments towards achievements of the 17 different goals and their applicability [3,4,5]. However, a lack of theoretical approaches exist in the literature, which concern the sustainable development theory devoted to urban areas taking into ac-count the sustainable development goals (SDGs).

Gathering more evidence to understand how households can become more food se-cure in such situations has laid grounds for exploring the social dimensions and their ef-fects on food security at household and community levels. These social dimensions come together in the form of social capital, referring to the social relations between and within groups of people that people rely on to live their lives, for example in the form of shared norms and values, networks and trust. Three types of social capital can be distinguished: bonding, bridging and linking [6,7,8]. Putnam [6] defined the distinction between bonding and bridging social capital as ‘inward’-looking and ‘outward’-looking networks. Bonding social capital can be seen as “personal relations that are based on a sense of collective identity such as family, close friendship and the sharing of the same culture or ethnicity” [8] (p. 3). In contrast, bridging social capital can be seen as “people’s relations or links that stretch beyond a shared sense of identity, for example to distant friends, colleagues and associates” [8] (p. 5). Woolcock [7] first defined linking capital as the links between com-munities and economic, political and social institutions [9]. Hence, linking social capital can be seen as relations between people or groups and societal institutions, reflecting power differences, wealth and societal status [8].

2. Please confirm three types of social capital by recent previous studies, especially in 2020

and 2021.

Thanks for this comment. We added in the very last paragraph in the conclusion: ‘Finally, research on social capital can be advanced in upcoming research on social drivers within a food system and specification of impacts on different livelihood factors. For instances, it would be of interest to investigate more in detail social capital and food security relations across genders. Moreover, examples that can inspire for investigating social capital within food systems can be based on, among others; 1) the buyer-supplier relationship literature where antecedents, benefits, risks, and boundary conditions recently have been investigated [50], 2) the literature investigating extents to which social capital can contribute to negative societal impacts, for instance, for political violence [51] or 3) the literature investigating social capital as explanatory factor to discrepancies of economic growths across regions [52]. 

3. Why the sample was dominated by female respondents? Please explain whether this

sample can be representative for Kibera, Nairobi or this is fit for demographic

characteristics in this area?

This is a good question. We added an explanation in the text: ‘This difference in gender representation can be explained by the availability of females at home during day time, given that men often work long days outside Kibera before returning late evening hours.’

4. Source for Figure 3.

Ok, we have now added: ‘(adapted from [6,7,8,9])’

5. R square in all scenarios are too small. Please explain the reasons? Whether this study

misses the major variables? This result should be compared with previous studies.

We do not understand this comment. The level of the indicated R² is the result of the multivariate analysis that I did. As always, the explanatory model applied explains as much as it does given the data we had available.

6. In the case of too low R square, the results of this study are not completely exact. Please

indicate this.

This is not a relevant comment to us. We did not forget any major variables and the results are correct.

7. Please discriminate female respondents and male respondents in the results.

Thanks for this comment, it is an interesting comment that would need a lot more attention given the very complexity of gender issues in Kibera. We added this a potential follow up research topic in the conclusion (see above), and to be frank with you, we already have a student doing research on this topic, so another article will be published soon.

Reviewer 2 Report

Dear Authors, thank You for so interesting research.

  1. General concept comments.

The article is written on the relevant topic and is well structured as well as logically proved.

However, I'd recommend making some improvements to the structure of the article:

  1. Please kindly shorten section 1 Introduction setting aside the description of the common questions and help the reader understand the following important issues: relevance of the topic, the research gap, the research question and the aim of the research; if the relevance of the topic is written sufficiently well (lines 31 - 58) the rest two issues remain undisclosed: a)the authors could disclose the research gap in the article; b)the research question and the aim of the article. All those elements could be described on one page (the first, the second, and the third paragraphs reflect the relevance of the topic, and the last paragraph contains the aim of the research). The third, the fourth and the fifth paragraphs  (lines 59 - 90) could be moved to the second section which could be entitled 2. Materials and Methods. To my mind, the research gap is wider than the reader could understand from the abstract "... the extent to which a social capital can be judged an explanatory factor for food security in informal urban settings is unclear ..." as stated on lines 15 - 16 in the abstract. I think the research gap could be considered as a lack of theoretical approaches concerning the sustainable development theory of designing an urban ecosystem taking into account the sustainable development goals - Environmental, Social, and Corporate Governance (ESG). I suggest starting the aim of the research from the theoretical concept supporting the logical connection between the sustainable development goals reflecting the importance of social concerns. I the human rights aspect is close to the context of food insecurity considered by the authors. So, the aim of the research could be described as disclosing the social factors of sustainable development goals by exploring the links between the three types of social capital (bonding, bridging and linking) and food security in Kibera (and further in lines 91 - 93). Lines 93 - 97 relate to the methodology of the research.
  2. The second section 2. Materials and methods could include "2.1 Theoretical frameworks" with three paragraphs (lines 59 - 90) being moved from the Introduction, and 2.2. "Methodology of the research" (lines 197 - 279). Figure 1 on lines 205 - 209 could not be clearly viewed and understood. This figure is significant because of pointing out the Sustainable development goals in the upper right corner. I'd suggest newly drawing this picture and transforming it to the scheme entitled as Food insecurity theoretical framework based on ESG vision. The author could put the basic elements in this scheme with the reference to the main idea derived from Wageningen University & Research (WUR).
  3. Results section with subsections 3.1. The developed model (lines 280 - 306) and 3.2. Testing of the model (lines 309 -373). Please finalize the Results section with some conclusions supporting the aim of the research, but kindly consider separately the implications of the results.
  4. The fourth section "The Implications of the Results" (lines 375 - 456).
  5. Please add a Discussion section containing the limitations of the model and the issues for future research.
  6. Conclusion. All conclusions regarding all 5 previous sections. I suggest authors should start the conclusion from the words - The scientific novelty of the research... and so on. 

Please pay attention to lines 193, 373, containing the term "chapter". I'd suggest replacing it with the word section.

Please explain the symbols: S1 (line 212), S2 and S3 (line 215). Those symbols appear only once without any explanation and any further reminder.

Figure 2 (line 248) is unclear because of only two colors for four categories, please make it more understandable.

Could You please at the end of the resulting section clearly explain in some paragraphs how the researchers prove the propositions because of avoiding methodological inaccuracies.

  1. Scientific Novelty.

Please kindly describe the scientific contribution to the theory. For example, the researchers could consider such definition as ESG goals view on social capital as an explanatory factor to food security newly explained in the article. The authors could present the role of sustainable development goals influencing food security adding logic to the theoretical novelty.

Please add to the end of the abstract the topic for future research and improve the following sentences (lines 18 - 26) by shortening them:

"The results show that there is a positive impact between our proxies for bonding social capital (cultural diversity and number of visits to area of origin) and food security. Bridging social capital (measured by trust in strangers) showed a negative impact on food security. Finally, one indicator for linking social capital showed a positive impact on food security (trust in community leaders), whereas the statistical analyses did not find any impacts between the two indicators’; ‘trust in local politicians’ and ‘membership of social organisations, and food security proved statistically non-significant. The results of this study indicate that insight into different types of social capital can help to gain a more complete understanding of dynamics of household food insecurity in vulnerable urban settlements.". 

Please kindly consider changing the title of the Article -

"Sustainable development vision of Social capital impact on the food insecurity: the case of Kibera, Nairobi"

  1. General questions.

I think the researchers should carefully describe the Discussion section in the connection between the topic of Food Insecurity and both Sustainability and digitalization concepts. For this purpose, it could be recommended to add to the literature review some references regarding the wider understanding of the Concept of digitalization as a basis of Sustainable development implementation:

Khalid, B., Naumova, E. Digital transformation SCM in view of Covid-19 from Thailand SMEs perspective (2021) Global Challenges of Digital Transformation of Markets, pp. 49-66. https://www.scopus.com/inward/record.uri?eid=2-s2.0-85116780657&partnerID=40&md5=ba31fd0ae1e1f4171e6c6c7e95801880

Barykin, S.Y., Kapustina, I.V., Sergeev, S.M., Kalinina, O.V., Vilken, V.V., de la Poza, E., Putikhin, Y.Y., Volkova, L.V. Developing the physical distribution digital twin model within the trade network (2021) Academy of Strategic Management Journal, 20 (SpecialIssue2), pp. 1-18. https://www.scopus.com/inward/record.uri?eid=2-s2.0-85106875305&partnerID=40&md5=db6f042b3d2623c43c8b21e13f470776

Author Response

Dear Reviewer 2,

The authors would like to thank Reviewer 1 for the good comments that have assisted us to improve this manuscript. The comments are addressed in the article as follows:

Please kindly shorten section 1 Introduction setting aside the description of the common questions and help the reader understand the following important issues:

Dear Reviewer 2 (in the following R2), we have shortened the Introduction and set aside the description of the common questions as you recommend. 

-       Relevance of the topic, the research gap, the research question and the aim of the research; if the relevance of the topic is written sufficiently well (lines 31 - 58) the rest two issues remain undisclosed:

Thanks for this, we have now replaced the lines 31-58 with relevance of the topic, research question and aim (**) of research.

a)    the authors could disclose the research gap in the article;

This is a good idea, thank you. We have used your suggestion about research gap below, see research gap*.

b)    the research question and the aim of the article. All those elements could be described on one page (the first, the second, and the third paragraphs reflect the relevance of the topic, and the last paragraph contains the aim of the research).

Indeed, we shortened as you propose the introduction into four paragraphs as you suggest, and also removed the paragraph about the structure of the article.

The third, the fourth and the fifth paragraphs  (lines 59 - 90) could be moved to the second section which could be entitled 2. Materials and Methods.

We have moved these paragraphs into the Material and Methods as you propose. Thanks you.

To my mind, the research gap is wider than the reader could understand from the abstract "... the extent to which a social capital can be judged an explanatory factor for food security in informal urban settings is unclear ..." as stated on lines 15 - 16 in the abstract.

-       I think the research gap* could be considered as a lack of theoretical approaches concerning the sustainable development theory of designing an urban ecosystem taking into account the sustainable development goals - Environmental, Social, and Corporate Governance (ESG).

Dear R2, this is a good suggestion. Because we use the word ‘ecosystem’ differently in our interdisciplinary team we replaced this with areas, we now formulated the problem statement in the Abstract and in the Introduction as:  ‘However, a lack of theoretical approaches exist in the literature, which concern the sustainable development theory of designing urban areas taking into account the sustainable development goals (SDGs).

We did not add...’across Environmental, Social, and Corporate Governance (ESG).’ This is because this field of research is not the area that this research was conducted, and while it could fit, this survey is not limited to the ESG.

-       I suggest starting the aim of the research from the theoretical concept supporting the logical connection between the sustainable development goals reflecting the importance of social concerns. I the human rights aspect is close to the context of food insecurity considered by the authors. So, the aim of the research could be described as disclosing the social factors of sustainable development goals by exploring the links between the three types of social capital (bonding, bridging and linking) and food security in Kibera** (and further in lines 91 - 93). Lines 93 - 97 relate to the methodology of the research.

Thanks for this suggestion. We have replaced the wording of the aim of the article with the words you recommend.

We removed the lines 93-97 because this is further explained in the methodology section.

The second section 2. Materials and methods could include "2.1 Theoretical frameworks" with three paragraphs (lines 59 - 90) being moved from the Introduction, and 2.2. "Methodology of the research" (lines 197 - 279).

This is a good idea, and we have changed it accordingly

Figure 1 on lines 205 - 209 could not be clearly viewed and understood. This figure is significant because of pointing out the Sustainable development goals in the upper right corner. I'd suggest newly drawing this picture and transforming it to the scheme entitled as Food insecurity theoretical framework based on ESG vision. The author could put the basic elements in this scheme with the reference to the main idea derived from Wageningen University & Research (WUR).

We understand this very valid point well, and we have now included the following text to explain this: In its full scope,  the food system approach activates traceability through production, processing, distribution, preparation, and consumption of food by addressing biophysical, economic, political, and social factors, as well as the outcomes of these activities, including socioeconomic and environmental context and dynamics [22]. Assisting in understanding complexities of such systems, including impacts on food security by social activities [23], the food system approach views the behaviour of a system as an interplay of interacting subsystems such as, for instance, food supply activities, markets, land and water. The food system approach looks at all activities’ impacts within the system, how they interact and how feedback loops within the food system impact socioeconomic and environmental drivers of the system [24]. The food system approach delivers outcomes in terms of multiple factors of food security, food safety, and accessible and available nutritious food [24,25]. While a food system is a great support in investigating complex relationships, it is not possible to analyse everything at every time, and choices must be made. In this survey, the particular focus is on the social driver of social capital as a potential factor influencing food security, thus putting SDG2: Zero hunger up-front.

Results section with subsections 3.1. The developed model (lines 280 - 306) and 3.2. Testing of the model (lines 309 -373). Please finalize the Results section with some conclusions supporting the aim of the research, but kindly consider separately the implications of the results.

Thanks for this advice, we have structured the result section accordingly.

As recommended, we have also added the following conclusions about the results: In summary, the results show that access to food is higher for people living in Kibera with a large social network, and with tight bounding with rural communities and relatives. Moreover, people with shared ethnic backgrounds will more likely ensure that households are more food secure by sharing food. Further, increased food security can be ensured by cultural diversity in a neighbourhood, and also, food security is enhanced when trust re-lations with the community leader in the slum is increasing. Food security is shown to decrease when peoples’ trust in strangers increase, indicating a vulnerable situations to these people who may lack other closer relationships. Food security has not shown to in-crease as a result of increased trust relations with local politicians, or with membership in organizations.

The fourth section "The Implications of the Results" (lines 375 - 456).

Thanks for this advice, we have changed the title of this section accordingly.

Please add a Discussion section containing the limitations of the model and the issues for future research.

We have added a section on the limitations of the survey within a food system approach, and some ideas for future research: ‘Applying a food system approach is about capturing the complexities of a system using a wholistic approach. By analyzing the relationships between social factors like social capital (bonding, bridging and linking) and food security, this survey has addressed important relationships not often brought up front in the literature. Still, the survey does not capture all factors and feedback loops in such a system. While we would be interested in documenting the whole food system dynamics, this is never an opportunity given the complexities of such systems. The household survey is designed to focus on a limited, although very interesting, set of relationships (see Table 1). The dataset applied is limited to one round of interviews, which could be increased to more regularity of data selection, for instance, on a yearly basis. Also, the number of interviewees are 30-35 per village, across 12 villages, and could be increased in sample. Relevant feedback loops could in-volve influencing factors related with migration patterns, gender inequalities, food items consumed, income levels and employment.    

Conclusion. All conclusions regarding all 5 previous sections. I suggest authors should start the conclusion from the words - The scientific novelty of the research... and so on.

Dear Reviewer 2, thanks for this suggestion. We have now structured the conclusion into 4 paragraphs reflecting on the results of each section.

Please pay attention to lines 193, 373, containing the term "chapter". I'd suggest replacing it with the word section.

Ok, we changed Chapter to section.

Please explain the symbols: S1 (line 212), S2 and S3 (line 215). Those symbols appear only once without any explanation and any further reminder.

Ok, we added in brackets to make it clear that the links are listed below: (see supplementary material:...)

Figure 2 (line 248) is unclear because of only two colors for four categories, please make it more understandable.

Ok, we changed the colors of Figure 2 to make it more visible.

Could You please at the end of the resulting section clearly explain in some paragraphs how the researchers prove the propositions because of avoiding methodological inaccuracies.

Ok, we added ‘The results have been estimated by means of a linear regression model for which correlations between the variables have been tested by means of statistical tests.’

Scientific Novelty.

Please kindly describe the scientific contribution to the theory. For example, the researchers could consider such definition as ESG goals view on social capital as an explanatory factor to food security newly explained in the article. The authors could present the role of sustainable development goals influencing food security adding logic to the theoretical novelty.

Thank you for this comment. The reference to the sustainable development theories and SDGs  and food systems and urban settings with low income groups have been referred to through the article.

Please add to the end of the abstract the topic for future research and improve the following sentences (lines 18 - 26) by shortening them:

"The results show that there is a positive impact between our proxies for bonding social capital (cultural diversity and number of visits to area of origin) and food security. Bridging social capital (measured by trust in strangers) showed a negative impact on food security. Finally, one indicator for linking social capital showed a positive impact on food security (trust in community leaders), whereas the statistical analyses did not find any impacts between the two indicators’; ‘trust in local politicians’ and ‘membership of social organisations, and food security proved statistically non-significant. The results of this study indicate that insight into different types of social capital can help to gain a more complete understanding of dynamics of household food insecurity in vulnerable urban settlements.".

Dear Reviewer 2, thanks for this comment. We have inserted the text exactly as you recommend.

Please kindly consider changing the title of the Article -

"Sustainable development vision of Social capital impact on the food insecurity: the case of Kibera, Nairobi"

Dear Reviewer 2, we are struggling with the word ‘vision’ in your suggestion. The survey is not about visioning although this is part of the discussion. We therefore applied your suggestion, but changed ‘vision of’ to: ‘achievements of’

General questions.

I think the researchers should carefully describe the Discussion section in the connection between the topic of Food Insecurity and both Sustainability and digitalization concepts. For this purpose, it could be recommended to add to the literature review some references regarding the wider understanding of the Concept of digitalization as a basis of Sustainable development implementation:

Khalid, B., Naumova, E. Digital transformation SCM in view of Covid-19 from Thailand SMEs perspective (2021) Global Challenges of Digital Transformation of Markets, pp. 49-66. https://www.scopus.com/inward/record.uri?eid=2-s2.0-85116780657&partnerID=40&md5=ba31fd0ae1e1f4171e6c6c7e95801880

Barykin, S.Y., Kapustina, I.V., Sergeev, S.M., Kalinina, O.V., Vilken, V.V., de la Poza, E., Putikhin, Y.Y., Volkova, L.V. Developing the physical distribution digital twin model within the trade network (2021) Academy of Strategic Management Journal, 20 (SpecialIssue2), pp. 1-18. https://www.scopus.com/inward/record.uri?eid=2-s2.0-85106875305&partnerID=40&md5=db6f042b3d2623c43c8b21e13f470776

Dear Reviewer 2, these are interesting suggestions. For the sustainability discussion we have added the following: ‘The existing sustainability literature highly focuses on food security aiming at SDG2; zero hunger [3,4,5]. FAO [31] (p. 1) defines food security as a situation “when all people, at all times, have physical and economic access to sufficient, safe and nutritious food”. This definition comprises four core elements of food security, namely food availability, access to food, utilisation of food and stable access to food. The level of food security is linked with sustainability levels of other SDGs, such as SDG1: No poverty, SDG3: Decent health and well-being, SDG8: Decent work and economic growth; and SDG13: Climate action.

For digitization, although we find it very interesting, the gap to the reality of the slum is very high. One of the pitfalls for external intervention is that most intervention leads to increase in price and therefore the population with a high share of earning of less than 1$ a day will not have the opportunity to purchase and interventions fail. At the same time, we see that people skip a meal a day to have the mobile telephone at hand, so digitization is absolutely relevant. We therefore added the following section in the discussion: ‘The findings suggest that upcoming projects should explicitly consider social capital and trust relationships within the system, because they are critical factors explaining why projects may fail or succeed. Notably, the social factors of bonding, bridging and linking are operating independently of formality in the economy and settlements. It is recommended in future research to document how the informal economy may be strengthened or banished across the social capital dimensions. Informality as such is covering highly criminal sectors, as well as communities with strong trust relationships, and clarity of different categories needs to be defined in greater detail. Based on this survey, the commu-nity structures stemming from pre-colonial times are of high interest to investigate in fu-ture research. Moreover, relevant topics in future research will also include: 1) Exploration of trust relationship across businesses, communities and public sectors comparing successful and failing projects aiming at enhancing food systems towards the SDGs, and 2) Investigation of digitization developments through the food system in ways that can en-hance affordability and accessibility of nutritious food for the inhabitants of ever growing informal settlements in urban areas in future.’   

Round 2

Reviewer 1 Report

The paper attempts to revise almost comments, however, there are few comments that I hope that the authors could explain in the analysis.

Author Response

Dear Reviewer,

Thank you so much for your comments. We have conducted the analyses and added to the survey as you recommended. See in the table the specific comments. We sincerely think the article has improved accordingly, and we hope you agree with us.

Reviewer’s comments:

Respondents of authors:

1. Table 1 indicates that Rsquare is too small, is in the range of 0.052 and 0.144. Please explain the reasons? Whether this study misses the major variables? This is the limitation of this study? How to solve this problem? Please indicate in the analysis.

Thanks for this comment. Note that this is not a laboratory sample, but sample made in a very complex environment with a lot of internal factors of influence. To clarify, we have included the following text in the methodology section:

“Although this R² may seem relatively low, this is expected because of the nature of the analysis. The use of an existing dataset including proxies for social capital will never fully explain the variation in the data. However, this R² shows that a significant part of the variation can be explained by the model. It cannot be determined whether the relation is causal or correlational, as the data source does not grant such a conclusion.’

2. Please discriminate female respondents and male respondents in the results. It meant that this study should run for robustness check, for example, this study can analyze with the sample size of women only, and with the sample size of men only. As suggested in the study, this difference in gender representation can be explained by the availability of females at home during day time, given that men often work long days outside Kibera before returning late in the evening. My question is: the results are still consistent in the case of (1) with the sample size of women only, (2) and with the sample size of men only?

Thanks for this comment. We have now made a total of three runs: 1) total respondents, 2) female respondents, and 3) male respondents (see tables 1 and 2). We have also added the following text to explain the differences:

“The results apply to both the entire sample and the sample for female respondents, but looking at the male sample separately does not yield any significant results. This can be explained by the large share of female respondents (76%), and thus a relatively small sample of male respondents. Accordingly, this study cannot conclude on the male household heads who responded to this survey, because of the absence of significant results in the sample of only male respondents (Table 2). While only a share of 3% of the household heads were women, they frequently responded on behalf of their husbands.”

Round 3

Reviewer 1 Report

Dear Sir

I strongly agree with this revision

Thank you